# Effects of primary health care and socioeconomic aspects on the dispersion of COVID-19 in the Brazilian Northeast: Ecological study of the first pandemic wave

**Luana Resende Cangussú**[1], **Jeisyane Acsa Santos Do Nascimento**[2], **Igor Rafael Pereira de Barros**[3], **Rafael Limeira Cavalcanti**[4], **Fábio Galvão Dantas**[5], **Diego Neves Araujo**[6], **José Felipe Costa da Silva**[2], **Thais Sousa Rodrigues Guedes**[2], **Matheus Rodrigues Lopes**[1], **Johnnatas Mikael Lopes**[1], **Marcello Barbosa Otoni Gonçalves Guedes**[2]*

1 Paulo Afonso Campus, Federal University of Vale do São Francisco, Paulo Afonso, Bahia, Brazil, 2 Campus Central, Federal University of Rio Grande do Norte, Natal, Rio Grande do Norte, Brazil, 3 Serra Talhada Campus, University of Pernambuco, Pernambuco, Brazil, 4 Capim Macio Campus, Uninassau Natal, Natal, Rio Grande do Norte, Brazil, 5 Central Campus, State University of Paraíba, Campina Grande, Paraíba, Brazil, 6 Federal University of Alagoas, Arapiraca, Alagoas, Brazil

* marcello.guedes@ufrn.br

**Data Availability Statement:** Below is the URL and DOI of the repository with the data available for

## Abstract

### Background

The COVID-19 pandemic has had a negative impact on socioeconomic and public health conditions of the population.

### Aim

To measure the temporal evolution of COVID-19 cases in cities near the countryside outside metropolitan areas of northeastern Brazil and the impact of the primary care organization in its containment.

### Methods

This is a time-series study, based on the first three months of COVID-19 incidence in northeastern Brazil. Secondary data were used, the outcome was number of COVID-19 cases. Independent variables were time, coverage and quality score of basic health services, and demographic, socioeconomic and social isolation variables. Generalizable Linear Models with first order autoregression were applied.

### Results

COVID-19 spreads heterogeneously in cities near the countryside of Northeastern Brazilian cities, showing associations with the city size, socioeconomic and organizational indicators of services. The Family Health Strategy seems to mitigate the speed of progression and

access. URL: https://doi.org/10.6084/m9.figshare.25122995.v1 DOI: 10.6084/m9.figshare.25122995.

**Funding:** The author(s) received no specific funding for this work.

**Competing interests:** The authors have declared that no competing interests exist.

burden of the disease, in addition to measures such as social isolation and closure of commercial activities.

## Conclusion

The spread of COVID-19 reveals multiple related factors, which require coordinated intersectoral actions in order to mitigate its problems, especially in biologically and socially vulnerable populations.

## Introduction

In December 2019, the first cases of coronavirus 2019 disease (COVID-19) occurred in China, caused by a new coronavirus called SARS-CoV-2 [1]. COVID-19 is the third recognized disease caused by the zoonotic coronavirus, preceded by severe acute respiratory syndrome (SARS) in 2002 and by Middle East respiratory syndrome (MERS) in 2012 [2].

When the virus was detected in different countries, the disease outbreak was declared by the World Health Organization (WHO) as an international public health emergency [3]. The rapid spread of COVID-19 demanded decisions and agile interventions by government agencies to minimize the impacts caused by the disease. Several strategies were implemented, such as closing schools, businesses, non-essential services and establishing lockdown. The essential aim was to increase the rate of social distancing, considered the most effective measure in controlling the spread of the disease, given the absence of specific therapies or vaccines [4].

In this complex scenario, COVID-19 has become an unprecedented challenge [5]. For instance, the SARS epidemic in 2002 caused approximately 800 deaths worldwide whereas the COVID-19 pandemic has caused a daily number of deaths that exceeded this level, with records of more than 1000 deaths per day in countries severely affected by the disease, such as China, Italy, Brazil and the United States [6].

The pandemic impacts are different in each country. In Brazil and other countries with lower socioeconomic development, for example, the disease is more intense, due to the low rate of social isolation, misinformation about basic health, little use of effective protective measures, inadequate housing and low-income conditions. For these reasons, unequal effects of the disease are evident in different locations [7, 8].

From this perspective, Primary Health Care (PHC) acts as a relevant instrument in the control of COVID-19 pandemic through the identification of vulnerabilities, service user testing, professional training, and monitoring of suspected and confirmed cases [9]. In an emergency situation such as the current one, PHC is essential to widely cover cases of the disease [10].

PHC assistance model implementation in Brazil is recent. Although it proves to be a promising proposal, it still presents structural, management and funding problems [11]. Despite this, PHC manages to act as a reality modifier in several places and social contexts, especially in small and medium-sized cities, where often a single basic family health staff is responsible for the care and epidemiological monitoring of an entire community [12].

In countries with continental dimensions, regional socioeconomic inequalities are huge [13]. In Brazil, large discrepancies and extensive regional gaps exist regarding the distribution of health services, especially in the north-northeast of the country [14].

Understanding the beginning of epidemic outbreak is necessary to manage the conditions of dissemination in order to be used as tools in the future for similar events. Thus, this study aims to measure the temporal evolution of COVID-19 cases in major cities outside

metropolitan regions of northeastern Brazil and the socioeconomic and organizational characteristics of the health system that may be related to the differences between them, according to the city size.

## Method

This is a population-based ecological time-series study with descriptive and analytical components, using secondary data. This design allows the establishment of the cause-effect relationship for public health interventions and social conditions as well as monitoring the evolution of large-scale events.

The investigated population was composed of residents of the largest towns near the countryside of northeastern Brazil, outside the metropolitan regions (Fig 1). The northeast region of Brazil encompasses the largest number of federative units (nine states), with approximately 53 million people, and an area equivalent to 18% of the national territory. Despite its relevance, this region is historically marked by great social inequality [11].

Data investigated in this time series corresponded to the first three months of COVID-19 incidence in Brazil until epidemiological week 22. The study outcome variable was diagnosed cases of COVID-19, which were treated as the incidence rate per 100,000 inhabitants. Secondary data were collected through the Health Surveillance System of the Ministry of Health (https://covid.saude.gov.br/), which are stored in the databases of the SUS IT Department (DATASUS).

The independent variables were time, analyzed by epidemiological weeks and ecological characteristics grouped into socioeconomic and health system dimensions. In the group of socioeconomic variables, it was used the Human Development Index (HDI), the Gini index, Gross Domestic Product (GDP) per capita, demographic density, the percentage of employed persons and the percentage of social isolation in the federative Units (https://cidades.ibge.gov.br/). In the variables of the health system, we used Family Health Strategy (FHS) coverage (https://egestorab.saude.gov.br/paginas/acessoPublico/relatorios/relHistoricoCobertura.xhtml) and the evaluation score of PHC health services quality by the Quality Monitoring and Evaluation Program (PMAQ score) (https://www.gov.br/saude/pt-br/composicao/saps/

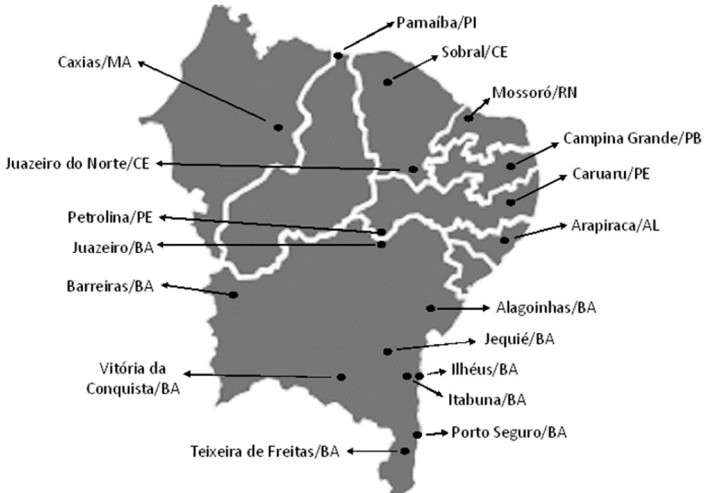

**Fig 1. Spatial demonstration of the largest cities in the interior of northeastern Brazil outside the metropolitan regions (adapted from https://earthobservatory.nasa.gov/map#5/-11.006/-43.989).**

pmaq). FHS coverage level was stratified into less than 50%, from 50 to 74% and above 75% of the population to estimate nested relationships with other variables [12]. The other variables were treated as they were collected.

The HDI is an important instrument for evaluating the development of certain locations. In Brazil, it is used as a key index of the United Nations millennium development goals. The Gini index was used as an instrument to measure the level of income concentration in the cities, used as a measure of social inequality, ranging from 0 to 1. The closer to 1, the greater the inequality in that location.

GDP per capita was used to assess the development of the municipalities (the higher its value, the greater the development of the city). Demographic density demonstrated the spatial distribution of inhabitants per square kilometer ($Km^2$). The percentage of employed persons included: those who worked at least one full hour with remuneration in money, products, goods or benefits (housing, food, clothing, training, etc.); those who worked without receiving direct remuneration (in aid to the economic activity of a member of the household or a relative who lives in another household); those who had paid work, but were temporarily away in the current week [15].

The social isolation index was extracted from the monitoring system of INLOCO website (https://mapabrasileirodacovid.inloco.com.br/pt/), which calculates the population level of social isolation. This index is estimated by the triangulation of cell towers, measuring the displacement of more than 200 meters of devices for personal use, as well as the agglomeration of these people in certain locations. For this purpose, polygons from all regions of the Brazilian Institute of Geography and Statistics (IBGE) database are used to ensure a more accurate categorization that represents reality.

The theoretical analysis model was set up to observe the association of temporal evolution of COVID-19 diagnoses, disease-associated deaths and the independent variables. These variables underwent crude analysis to estimate their relationship with the outcomes, and they were later included in an adjusted model to extract the main effects of each factor. Categorical variables also performed a nested analysis, based on their conceptual and spatial relationship, if they had a main effect on the adjusted model.

Statistical analysis was performed using Generalizable Linear Models (GLM) with an auto-regressive work correlation matrix of order 1, as it is a time-series where previous cases affect the number of subsequent cases. The log ligand function with Gamma distribution was used, as the data from the independent variables did not show a direct linear relationship with the outcome. The best fit of the model was established by the Quasi-likelihood under Independence Model Criterion (QIC), where lower values refer to models that are more adjusted to the data and help to organize the variables in the adjusted model. The analysis was stratified for cities with up to 300 thousand inhabitants (medium-sized) and more than 300 thousand inhabitants (medium-large), according to IBGE classification.

Hypothesis tests were performed using Wald's chi-square test between the outcome variable and the independent variables, selecting those with "p" values equal to or less than 0.10 to be included in the adjusted model. A significance level of $\leqq 0.05$ was considered. Additionally, the coefficients (B) of the raw and adjusted model were used as effect measures to present the magnitude of the relationship between variables.

## Results

Eighteen medium and medium-large cities were analyzed, whose sociodemographic and PHC organizational characteristics are shown in Table 1. There is a large variation in GDP per capita (±3339.70) and a homogeneous but high Gini index, which reflects a considerable income

**Table 1. Diagnosed cases of COVID-19, deaths, lethality up to the 22nd epidemiological week in the eighteen largest cities in the interior of northeastern Brazil outside the metropolitan regions.**

| County | Accumulated cases | Death cases | Population | Size | Demographic density | Poverty Incidence | FHS coverage (%) | PMAQ score | GDP Per capita | %Occupied people | Gini | HDI |
|---|---|---|---|---|---|---|---|---|---|---|---|---|
| Caxias/MA | 96 | 3 | 164880 | Medium | 30.12 | 58.44 | 100 | 0.00 | 10538.06 | 10.00 | 0.43 | 0.62 |
| Sobral/CE | 1126 | 37 | 208935 | Medium | 88.67 | 49.30 | 100 | 3.23 | 21679.33 | 24.60 | 0.47 | 0.71 |
| Porto Seguro/BA | 90 | 1 | 148686 | Medium | 52.70 | 52.17 | 99.77 | 2.60 | 18888.98 | 22.60 | 0.47 | 0.68 |
| Arapiraca/AL | 323 | 7 | 231747 | Medium | 600.83 | 60.44 | 99.74 | 2.01 | 17511.69 | 17.30 | 0.43 | 0.65 |
| Juazeiro do Norte/CE | 106 | 9 | 274207 | Medium | 1004.45 | 52.14 | 99.40 | 3.23 | 16375.01 | 20.00 | 0.46 | 0.69 |
| Parnaíba/PI | 223 | 6 | 153078 | Medium | 334.51 | 54.01 | 99.17 | 2.00 | 13534.25 | 14.80 | 0.43 | 0.69 |
| Juazeiro/BA | 36 | 3 | 216707 | Medium | 30.45 | 45.24 | 93.93 | 2.53 | 16687.70 | 17.20 | 0.49 | 0.68 |
| Campina Grande/PB | 1288 | 10 | 409731 | Medium-Large | 648.31 | 58.88 | 91.78 | 2.40 | 21077.30 | 26.90 | 0.45 | 0.72 |
| Petrolina/PE | 143 | 5 | 349145 | Medium-Large | 64.44 | 42.96 | 88.93 | 2.35 | 17454.51 | 19.20 | 0.46 | 0.70 |
| Teixeira de Freitas/BA | 86 | 0 | 160487 | Medium | 118.87 | 53.01 | 85.99 | 3.35 | 14298.26 | 16.70 | 0.46 | 0.68 |
| Mossoró/RN | 986 | 36 | 297378 | Medium | 123.76 | 55.28 | 75.41 | 2.83 | 20858.33 | 22.20 | 0.46 | 0.72 |
| Caruaru/PE | 335 | 21 | 361118 | Medium-Large | 342.07 | 33.69 | 72.61 | 3.65 | 19311.06 | 22.70 | 0.44 | 0.68 |
| Alagoinhas/BA | 59 | 1 | 151596 | Medium | 188.67 | 39.84 | 70.55 | 2.37 | 22500.08 | 16.90 | 0.46 | 0.68 |
| Itabuna/BA | 884 | 3 | 213223 | Medium | 473.50 | 42.83 | 67.96 | 2.00 | 18023.73 | 21.40 | 0.50 | 0.71 |
| Jequié/BA | 349 | 6 | 155966 | Medium | 47.07 | 48.95 | 66.36 | 1.74 | 15765.90 | 16.20 | 0.49 | 0.66 |
| Barreiras/BA | 49 | 0 | 155439 | Medium | 17.49 | 40.90 | 64.37 | 2.41 | 24676.48 | 19.90 | 0.50 | 0.72 |
| Ilhéus/BA | 523 | 9 | 162327 | Medium | 104.67 | 47.34 | 48.88 | .66 | 21789.59 | 19.40 | 0.50 | 0.69 |
| Vitória da Conquista/BA | 105 | 4 | 338480 | Medium-Large | 91.41 | 39.06 | 46.46 | 3.35 | 18589.99 | 22.20 | 0.47 | 0.68 |

Medium size (100–300 thousand), medium-large size (300–500 thousand); FHS: Family Health Strategy; GDP: Gross Domestic Product; HDI: Human Development Index

concentration. The analyzed cities also have a heterogeneous demographic density (±266.79), which may reflect wide opportunities for unsafe contacts. Poverty affects a third of the inhabitants, and some cities have more than 60% of poor inhabitants.

There was a general exponential growth of COVID-19 cases ($y = 0.00278^{0.49x}$). However, the pattern is not identical for all cities, with four "levels" in the progression curve (Fig 2).

Among the medium-sized cities, those with the most COVID 19 cases were Sobral/CE, Itabuna/BA and Mossoró/RN. Among the medium-large cities, Campina Grande/PB was the most affected. FHS coverage was between 40% and 100%. From all investigated cities, four were medium-large cities (Table 1).

Analyzing the cities characteristics that may be associated with COVID-19 dissemination, the following variables were identified in the raw model: GDP per capita, the percentage of occupied persons, Gini index, HDI, FHS coverage, the total time of closed commercial activities and evolution of epidemiological weeks (Table 2).

In the adjusted model, it was observed that, in medium-sized cities, a FHS coverage rate of less than 75% of the population is associated with sixfold more COVID-19 cases ($B_{<50\%} = 1.79$ and $B_{50-74} = 1.81$). The increase in the poverty incidence ($B = 0.16$) and the percentage of

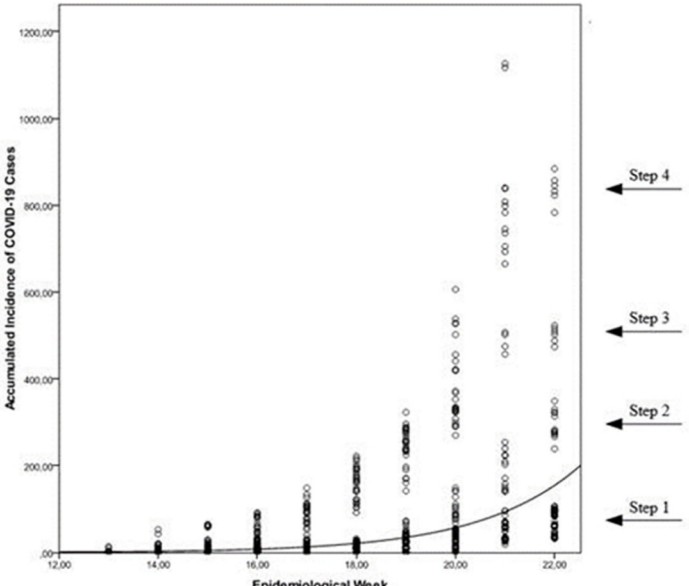

**Fig 2. Temporal distribution of COVID-19 cases in the largest cities in the interior of Northeast Brazil outside the metropolitan regions.**

occupied persons (B = 0.17) were also related to a higher incidence of contagion. On the other hand, the increase in social isolation (B = -0.010) and time of closed commercial activities (B = 0.016) had an inverse relationship with the increase in the incidence of cases (Table 3).

In medium-large cities, FHS coverage above 75% of the population was related to a higher COVID-19incidence, when compared to smaller ones. This paradoxical scenario may be explained by the interaction between FHS coverage and PMAQ score: the only city with more than 75% has a quality assessment with a score of less than 3 (Petrolina/PE), whereas cities with coverage <50% and 50–74% have grades higher than 3. Similarly, the capacity for social isolation resulted in an inverse relationship regarding COVID-19 cases (B = -6.17; p<0.001), reinforcing the effect of the measures by local governments.

In contrast, the analysis of occupied people (B = 0.09; p<0.001) showed a directly proportional relationship: the more people developed productive occupational activities in cities, the more diagnosed COVID-19 cases were confirmed. On the other hand, the time of closed commercial activities did not show a significant association with the outcome in cities of this size (Table 3). HDI and Gini sociodemographic variables showed strong multicollinearity with poverty incidence, which seems to better explain this outcome compared to the others. Demographic density was also removed from the adjusted model as it produces higher QIC and, therefore, a lower global adjustment.

## Discussion

It was observed that COVID-19 was heterogeneous in its distribution throughout the cities near the countryside of the Brazilian Northeast before the spread was high. In addition to the cities size and their geographic position, it is very likely that contextual social, economic and health services organization characteristics interact with the speed of virus dissemination.

Traditionally, Brazil is marked by contrasting regional differences inherited from a historical past of land and income concentration. These disparities resulted in large socioeconomic,

**Table 2. Crude analysis for the association between contextual factors and the occurrence of COVID-19 cases in the largest medium-sized cities in the interior of Northeast Brazil outside the metropolitan regions.**

| Parameter | B | SE | Wald's 95% Confidence Interval | | Hypothesis test | |
|---|---|---|---|---|---|---|
| | | | Bottom | Higher | $x^2$ | P |
| **medium size cities** | | | | | | |
| *per capita* GDP | <0.001 | 4.9553E-5 | 3.837E-5 | <0.001 | 7.48 | 0.006 |
| Demographic density | <0.001 | <0.001 | -0.002 | 0.001 | 0.07 | 0.80 |
| Occupied people | 0.17 | 0.05 | 0.06 | 0.27 | 9.48 | 0.002 |
| GINI | 14.90 | 7.19 | 0.81 | 29.00 | 4.30 | 0.04 |
| HDI | 21.95 | 7.48 | 7.30 | 36.61 | 8.62 | 0.003 |
| poverty index | -0.006 | 0.05 | -0.10 | 0.09 | 0.02 | 0.90 |
| FHS coverage | | | | | | |
| <50% | 0.97 | 0.40 | 0.19 | 1.7 | 5.95 | 0.01 |
| 50–74% | 0.06 | 0.74 | -1.38 | 1.51 | 0.008 | 0.93 |
| ≥ 75% | 0 | | | | | |
| PMAQ score | -0.07 | 0.30 | -0.65 | 0.51 | 0.06 | 0.81 |
| Mask time | 0.003 | 0.004 | -0.01 | 0.01 | 0.46 | 0.50 |
| Social isolation | 0.001 | <0.001 | <0.001 | 0.002 | 2.65 | 0.10 |
| Epidemiological week | 0.16 | 0.05 | 0.05 | 0.26 | 9.13 | 0.003 |
| Total closed trade time (days) | -0.02 | 0.008 | -0.04 | -0.009 | 10.29 | 0.001 |
| **medium-large cities** | | | | | | |
| *per capita* GDP | 0.0002 | 4.97E-5 | 0.001 | 0.003 | 31.30 | <0.001 |
| Demographic density | 0.002 | 0.0001 | 0.001 | 0.002 | 211.94 | <0.001 |
| Occupied people | 0.11 | 0.026 | 0.06 | 0.16 | 17.57 | <0.001 |
| GINI | -20.23 | 11.55 | -42.88 | 2.42 | 3.06 | 0.08 |
| HDI | 17.17 | 5.05 | 7.26 | 27.09 | 11.53 | 0.001 |
| Poverty index | 0.03 | 0.01 | 0.01 | 0.05 | 8.54 | 0.003 |
| FHS coverage | | | | | | |
| <50% | -0.57 | 0.30 | -1.16 | 0.01 | 3.59 | 0.05 |
| 50–74% | -0.28 | 0.30 | -0.88 | 0.30 | 0.89 | 0.34 |
| ≥ 75% | | | | | | |
| PMAQ score | -0.30 | 0.28 | -0.86 | 0.26 | 1.11 | 0.29 |
| Mask time | 0.003 | 0.02 | -0.04 | 0.04 | 0.01 | 0.89 |
| Social isolation | -0.004 | 0.0006 | -0.005 | -0.003 | 40.32 | <0.001 |
| Epidemiological week | 0.60 | 0.06 | 0.46 | 0.74 | 76.29 | <0.001 |
| Total closed trade time (days) | 0.01 | 0.005 | 0.008 | 0.03 | 11.34 | 0.001 |

FHS: Family health strategy; B: Coefficient of the equation; SE: Standard error; 95% CI: 95% confidence interval; $x^2$: Wald's chi-square; p: test probability; E-5: Scientific notation with base 10 and exponent -5.

educational and health differences [16]. Currently, the South and Southeast regions of the country concentrate most of the ICU beds, respirators, and other complex services, whereas in the Northeast and North, there is a notable lack or absence of these devices, which may explain higher mortality/100,000 inhabitants by COVID-19 in these regions [17]. This scenario highlights the need for better organization of the health services, for better regional planning and ensuring adequate transport to reference centers when necessary [18].

Due to its continental dimensions, Brazil has also intra-regional discrepancies. This study evidenced, for example, marked differences existing within the northeast region, with variations in GDP per capita, HDI, GINI, PHC and FHS coverage. This heterogeneity reveals traces

**Table 3. Adjusted model for the association between contextual factors and the occurrence of COVID-19 cases, stratified by medium and medium-large size of the largest cities in the interior of Northeast Brazil outside the metropolitan regions.**

| Parameter | B | SE | CI95% | | hypothesis test | | | QICC |
|---|---|---|---|---|---|---|---|---|
| | | | Lower | Higher | $x^2$ | df | P | |
| **medium size cities** | | | | | | | | 878.00 |
| FHS coverage (%) | | | | | | | | |
| <50% | 1.80 | 0.17 | 1.46 | 2.13 | 109.82 | 1 | <0.001 | |
| 50–74% | 1.82 | 0.40 | 1.03 | 2.60 | 20.69 | 1 | <0.001 | |
| ≥ 75% | 0 | | | | | | | |
| Epidemiological week | 0.21 | 0.04 | 0.13 | 0.29 | 25.76 | 1 | <0.001 | |
| Occupied people (%) | 0.18 | 0.03 | 0.12 | 0.23 | 33.60 | 1 | <0.001 | |
| Poverty index | 0.17 | 0.02 | 0.13 | 0.20 | 71.35 | 1 | <0.001 | |
| Social isolation (%) | -0.01 | 0.002 | -0.01 | -0.007 | 32.46 | 1 | <0.001 | |
| Total closed trade time (days) | -0.02 | 0.006 | -0.03 | -0.003 | 6.35 | 1 | 0.012 | |
| Scale | 1.329 | | | | | | | |
| **medium-large cities** | | | | | | | | 1169.22 |
| Social isolation (%) | -6.17 | 0.974 | -8.08 | -4.26 | 40.19 | 1 | <0.001 | |
| FHS coverage (%) | 0.01 | 0.003 | 0.008 | 0.02 | 18.71 | 1 | <0.001 | |
| <50% | -0.08 | 0.047 | -0.18 | 0.007 | 3.25 | 1 | 0.07 | |
| 50–74% | -0.16 | 0.069 | -0.29 | -0.02 | 5.40 | 1 | 0.02 | |
| ≥ 75% | 0 | | | | | | | |
| Occupied people (%) | 0.09 | 0.013 | 0.06 | 0.11 | 48.46 | 1 | <0.001 | |
| Total closed trade time (days) | -0.004 | 0.003 | -0.01 | 0.003 | 1.29 | 1 | 0.25 | |
| Epidemiological week | 0.44 | 0.039 | 0.36 | 0.51 | 122.40 | 1 | <0.001 | |
| Scale | 0.27 | | | | | | | |

FHS: Family Health Strategy; B: Coefficient of the equation; SE: Standard Error; CI95%: 95% Confidence interval; $x^2$ Wald's chi-square; df: degrees of freedom p:test probability.

of an unequal developmental process, with resources initially concentrated in states/counties such as Pernambuco, Ceará and Bahia, from investments by the Northeast Development Superintendence (Sudene) in the 1960s [19]. The existence of such great intraregional variability may represent difficulties in accessing health services and also conditions of precarious care in some locations, which becomes an impasse in pandemic times [20].

Faced with a highly infective virus, the disease spread in Brazil would not be different from what was seen in China and Italy. Community transmission in some parts of the country started less than a month after the detection of the first case and, since then, the number of infected people has grown exponentially, as observed in our study, which denoted the growth in the number of cases in the largest northeastern cities outside the metropolitan area [21].

In addition, the study showed the number of cases accumulated in medium-sized cities, especially in Sobral/CE, Itabuna/BA and Mossoró/RN. Until the time of this study, these cities registered the largest number of cases, which may be explained because both cities are located on routes with large national road traffic, facilitating the spread of the disease.

It was also observed that certain characteristics of medium-sized cities may be associated with COVID-19 dissemination, such as a high poverty incidence and income concentration and HDI below the national average. Among the medium-large cities, only Campina Grande-PB had records inconsistent with COVID-19, but proportionally lower than the other cities. Its geographical position close to the coast, with fast connection to the state capital, as a major road circulation route can be contributing aspects. Historically, Campina Grande-PB is a city on the

route of goods and travelers, and it is in the center of a metropolitan area that is home for more than 1.5 million inhabitants, who circulate daily in its urban area to work, to obtain services or to go shopping. The city is also served by air connection to large cities such as Recife/PE and São Paulo/SP, which may have increased the virus circulation by air transportation.

The cities size, theoretically, influences the magnitude of socioeconomic indicators and organization of the local health system. Larger cities present a more homogeneous inequality along with more wealth; the medium-sized ones are more heterogeneous regarding these indicators, interfering with the uneven pandemic effects [22, 23].

Social distancing, for example, is often impractical for residents who provide essential services, who need to work daily, often traveling by public transportation [24, 25]. While home office is a possibility for some people, for others, the exposure to unhealthy working conditions is the only feasible possibility [26]. Other aspects make it difficult to achieve and maintain social isolation, such as precarious housing situations, violent housing environments, lack of financial assistance, low health education and the work environment [4, 27].

It was also observed that the time of closure of commercial activities was inversely related to increase in the incidence of cases in medium-sized cities, without a similar association in medium-large cities. It is inferred that the pressure from the business sector is stronger in larger cities, given that its impact on the local economy encompasses a greater proportion of tax revenue. Closing commercial establishments is an effective action to flatten the COVID-19 progression curve, as it reduces the circulation of people and agglomerations. However, this is a complex measure to apply and maintain, given the great economic impact caused and the pressure from different sectors of the economy to return to their activities. Some authors even compare the current economic situation to the scenario experienced during World War II [28].

The impact caused by the pandemic and the adoption of several measures to contain the disease spread was not only reflected in Health and Economics, but also highlighted political and psychosocial nuances [29–31]. The COVID-19 pandemic exposed the vulnerability of the health systems in several countries, inefficient to respond with adequate quality to the installed public health emergency situation [9].

Since the promulgation of the Federal Constitution and the Organic Health Laws (Law No. 8080), which guide the functioning of the Unified Health System (SUS), Brazil still faces numerous difficulties in implementing all the aspects that underlie this system. The cause/consequence of this problem is the lack of proportionality regarding the binomial of the population's health needs and equitable access to an adequate service network, so that there is comprehensive care [15, 30, 32].

Despite the findings of great interest for crisis management and planning of future protective policies, there are limitations to be exposed. The first one refers to the use of non-recent socioeconomic indicators. However, it is believed that such estimates have regressed due to the economic crisis in Brazil, implying a worse scenario. The second limitation is that the low number of observations in medium-large cities can interfere with the variability of data for the construction of inferences when this characteristic is considered. The last limitation refers to the use of the social isolation indicator collected for the entire unit of the federation, which is, therefore, a proxy for social isolation in cities. However, this variable had statistical significance and may reveal more impact if local estimates are applied.

## Conclusion

The COVID-19 pandemic has shown great differences in several countries, and within them, inter-regional differences, directly reflecting characteristics of health systems and population contextual factors.

In Brazil, the profound socioeconomic and cultural imbalance, resulting from centuries of a mistaken political attitude, is directly reflected in the heterogeneity of health care, a fact that becomes even more relevant in a serious health emergency, such as the COVID-19 pandemic.

In the Brazilian Northeast, despite the shortages in secondary and tertiary health services, PHC has taken a leading role in monitoring cases and substantially contributing to minimize the catastrophic effects that spread in this critical period experienced by the world. In addition, timely mitigating social inequalities may generate less inequities in the manifestation of COVID-19 in the population and, probably, in other harmful health conditions.

## Supporting information

**S1 Data.**
(XLSX)

## Author Contributions

**Data curation:** Luana Resende Cangussú.

**Formal analysis:** Igor Rafael Pereira de Barros.

**Funding acquisition:** Rafael Limeira Cavalcanti, Fábio Galvão Dantas.

**Investigation:** José Felipe Costa da Silva, Thais Sousa Rodrigues Guedes, Matheus Rodrigues Lopes.

**Methodology:** Jeisyane Acsa Santos Do Nascimento, Diego Neves Araujo.

**Project administration:** Johnnatas Mikael Lopes.

**Supervision:** Marcello Barbosa Otoni Gonçalves Guedes.

**Writing – review & editing:** Johnnatas Mikael Lopes, Marcello Barbosa Otoni Gonçalves Guedes.

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
