## [Decision Letter · Decision Letter 0]

27 Sep 2023

PONE-D-23-12150Effects of Primary Health Care and Socioeconomic Aspects on the Dispersion of COVID-19 in the Brazilian Northeast: Ecological Study of the First Pandemic WavePLOS ONE

Dear Dr. Costa da Silva,

Thank you for submitting your manuscript to PLOS ONE. After careful consideration, we feel that it has merit but does not fully meet PLOS ONE’s publication criteria as it currently stands. Therefore, we invite you to submit a revised version of the manuscript that addresses the points raised during the review process.

We look forward to receiving your revised manuscript.

Kind regards,

Raphael Mendonça Guimaraes, PhD

Academic Editor

PLOS ONE

Journal Requirements:

2. We note that Figure 01 in your submission contain map images which may be copyrighted. All PLOS content is published under the Creative Commons Attribution License (CC BY 4.0), which means that the manuscript, images, and Supporting Information files will be freely available online, and any third party is permitted to access, download, copy, distribute, and use these materials in any way, even commercially, with proper attribution. For these reasons, we cannot publish previously copyrighted maps or satellite images created using proprietary data, such as Google software (Google Maps, Street View, and Earth). For more information, see our copyright guidelines: http://journals.plos.org/plosone/s/licenses-and-copyright.

               1. You may seek permission from the original copyright holder of Figure01 to publish the content specifically under the CC BY 4.0 license. 

Reviewers' comments:

Reviewer's Responses to Questions

**Comments to the Author**

1. Is the manuscript technically sound, and do the data support the conclusions?

Reviewer #1: Yes

Reviewer #2: Partly

2. Has the statistical analysis been performed appropriately and rigorously? 

Reviewer #1: I Don't Know

Reviewer #2: No

3. Have the authors made all data underlying the findings in their manuscript fully available?

Reviewer #1: No

Reviewer #2: Yes

4. Is the manuscript presented in an intelligible fashion and written in standard English?

Reviewer #1: No

Reviewer #2: No

5. Review Comments to the Author

Reviewer #1: I want to congratulate the authors of the study. They have addressed a relevant topic and proposed a sound analysis methodology. I have some comments about some sections which could be improved.

1 - The Results section is somewhat confusing. The word "capta" in line 195 is misspelt. The figure after line 202 is in Portuguese. The word "closed" in line 214 is also misspelt. There should be a more clear distinction when the author is referring to medium-sized cities and medium-large cities because sometimes it is not clear. There are some discussions about the Results that would fit better in the appropriate section ("Discussion").

2 - In the discussion section, the author should revise their main findings and outline them to the reader before proceeding to the discussion itself. It would make the text clearer to understand. In line 246, the author mentions the lethality of the virus, although it is not a variable assessed in the study.

3 - In line 275, the sentence " Furthermore, the decisions in public health may have been wrong for the population needs" needs further explanation. It is not clear in the text previously what the author is referring to.

Reviewer #2: The study evaluates a relevant aspect of the Covid-19 pandemic.

I have doubts whether, at this point, the analysis carried out is relevant, given that this is the initial phase of the pandemic, with many cases still underreported or poorly classified as COVID-19. This should at least be included as a limitation of the study, and I recommend that the introduction mention why this analysis still contributes to the scientific community.

Contrary to what the authors say, it is incorrect to say that "this design allows the establishment of the cause-effect relationship for public health interventions and social conditions as well as monitoring the evolution of large-scale events." Although analytical, the nature of the ecological study does not allow for establishing a causal relationship. Although it is an ecological study, the authors say that the outcome variable is people diagnosed with Covid-19. This is not correct. If the survey is ecological (as it appears to be), the variable is the rate of each city analyzed, compared to the CITIES' indicators (and not the people's, as mentioned).

It was not clear how the time series was modeled. What was in association? Is the rate of increase in Covid-19 cases? The volume of cases during the period? How can we explain the association with variables that historically do not have a normal distribution, such as income indicators?

Although the data looks interesting, the authors made the interpretation very superficial. Apparently, for example, the size of cities is an effect modifier for income inequality and development indicators. What explains this effect? Is there plausibility for this? Did the authors test models with interaction between these variables?

Considering the tiny n (significantly when stratified by the size of the cities), what was the quality of fit of the multivariate model? I want this analysis to be included in the results.

Finally, the figure and table contain words without translation into English.

For others, the translation is precarious ("busy people" instead of "occupied people"; use of "," as a decimal divider, etc.). I strongly recommend an extensive review of the translation of the article.

6. PLOS authors have the option to publish the peer review history of their article (what does this mean?). If published, this will include your full peer review and any attached files.

Reviewer #1: No

Reviewer #2: No

---

## [Author Response · Author response to Decision Letter 0]

22 Nov 2023

September 29, 2023

PLOS ONE

Response to reviewers

PONE-D-23-12150

Effects of Primary Health Care and Socioeconomic Aspects on the Dispersion of COVID-19 in the Brazilian Northeast: Ecological Study of the First Pandemic Wave

2. We note that Figure 01 in your submission contain map images which may be copyrighted. All PLOS content is published under the Creative Commons Attribution License (CC BY 4.0), which means that the manuscript, images, and Supporting Information files will be freely available online, and any third party is permitted to access, download, copy, distribute, and use these materials in any way, even commercially, with proper attribution. For these reasons, we cannot publish previously copyrighted maps or satellite images created using proprietary data, such as Google software (Google Maps, Street View, and Earth)

Reply: Edited figure based on suggested domain NASA Earth Observatory (public domain): http://earthobservatory.nasa.gov/. Figure created from the following link: 

https://earthobservatory.nasa.gov/map#5/-11.006/-43.989

Reply: Dear editorial team, on line 139, in the methods section, the electronic address of the database for the outcomes is described. The database for the independent variables was specified in the text in its new version.

Review Comments to the Author

Reviewer 1

1 - The Results section is somewhat confusing. The word "capta" in line 195 is misspelt. The figure after line 202 is in Portuguese. The word "closed" in line 214 is also misspelt. There should be a more clear distinction when the author is referring to medium-sized cities and medium-large cities because sometimes it is not clear. There are some discussions about the Results that would fit better in the appropriate section ("Discussion"). 

Reply: The suggestions were accepted.

2 - In the discussion section, the author should revise their main findings and outline them to the reader before proceeding to the discussion itself. It would make the text clearer to understand. In line 246, the author mentions the lethality of the virus, although it is not a variable assessed in the study.

Reply: The suggestions were accepted.

3 - In line 275, the sentence " Furthermore, the decisions in public health may have been wrong for the population needs" needs further explanation. It is not clear in the text previously what the author is referring to.

Reply: The suggestions were accepted.

Reviewer 2 

The study evaluates a relevant aspect of the Covid-19 pandemic.

I have doubts whether, at this point, the analysis carried out is relevant, given that this is the initial phase of the pandemic, with many cases still underreported or poorly classified as COVID-19. This should at least be included as a limitation of the study, and I recommend that the introduction mention why this analysis still contributes to the scientific community.

Reply: The suggestions were accepted: “Understanding the beginning of epidemic outbreaks is necessary to manage the conditions of dissemination in order to become tools in the future for similar events.”

Contrary to what the authors say, it is incorrect to say that "this design allows the establishment of the cause-effect relationship for public health interventions and social conditions as well as monitoring the evolution of large-scale events." Although analytical, the nature of the ecological study does not allow for establishing a causal relationship. Although it is an ecological study, the authors say that the outcome variable is people diagnosed with Covid-19. This is not correct. If the survey is ecological (as it appears to be), the variable is the rate of each city analyzed, compared to the CITIES' indicators (and not the people's, as mentioned).

Reply: We agree with the analysis that the outcome of this study is the incidence rate of COVID-19 diagnoses.

However, despite the ecological aspect, we emphasize that time series designs are useful in estimating causal effects when it is not possible to apply classic designs such as clinical trials or cohorts.

(Jiang, H., Feng, X., Lange, S. et al. Estimating effects of health policy interventions using interrupted time-series analyses: a simulation study. BMC Med Res Methodol 22, 235 (2022). https://doi.org/10.1186/s12874-022-01716-4

Pinto R., Valentim R, Silva LF, Souza G F. ,Use of Interrupted Time Series Analysis in Understanding the Course of the Congenital Syphilis Epidemic in Brazil. The Lancet 7 (2022). DOI:https://doi.org/10.1016/j.lana.2021.100163

It was not clear how the time series was modeled. What was in association? Is the rate of increase in Covid-19 cases? The volume of cases during the period? How can we explain the association with variables that historically do not have a normal distribution, such as income indicators?

Reply: The time series was modeled using GEE Analysis, which applies to situations of repeated measurements where the outcome does not present a normal distribution and the data are correlated, as specified in the methods section. It is not necessary to have a normal distribution of the outcome to produce modeling. In fact, it is unusual for health outcomes to present a normal distribution when it comes to health indicators. Therefore, it is necessary to use other families of exponential distribution to carry out the modeling.

The association investigated in the research is explained in the theoretical model proposed for data analysis, aiming to estimate changes in the evolution of rates through interference from independent variables.

Although the data looks interesting, the authors made the interpretation very superficial. Apparently, for example, the size of cities is an effect modifier for income inequality and development indicators. What explains this effect? Is there plausibility for this? Did the authors test models with interaction between these variables?

Considering the tiny n (significantly when stratified by the size of the cities), what was the quality of fit of the multivariate model? I want this analysis to be included in the results.

Reply: 

We understand that the statement that the size of cities is an effect modifier of social inequality and development is simplistic and, therefore, mistaken. This relationship is not linear, even when selecting a sample of cities in the same population size stratum as established by the Brazilian Institute of Geography and Statistics.

The statement disregards factors such as the geographical location of cities and their history of social and economic development, especially in the Brazilian northeast where there is an imbalance between coastal cities and the hinterland, for example.

We can highlight that stratification by population size helps in better analysis and modeling, where there is also control over population density, a characteristic that strongly affects the outcome as it is influenced by contact between people.

Regarding the interaction test between the independent variables in the model, it was not the focus of our analysis, only the independent effect, as there was no evidence of multicollinearity in the model quality control analyses. This can be analyzed in other investigations.

Quality of fit was assessed using the Quasi-likelihood under Independence Model Criterion (QIC), whose value has no interpretation of practical applicability in public health, only lower model values reveal better fit. For this reason, it was omitted from the results because it is an internal quality assessment statistic and any scientist can audit the model using the same database. However, we updated table 3 and included the QIC value for each model.

Including excessive quality control statistics on freely available data would make the text uninteresting to read.

Finally, the figure and table contain words without translation into English.

For others, the translation is precarious ("busy people" instead of "occupied people"; use of "," as a decimal divider, etc.). I strongly recommend an extensive review of the translation of the article.

---

## [Editor Report · Decision Letter 1]

20 Dec 2023

Efeitos da Atenção Primária à Saúde e dos Aspectos Socioeconômicos na Dispersão da COVID-19 no Nordeste Brasileiro: Estudo Ecológico da Primeira Onda Pandêmica

PONE-D-23-12150R1

Dear Dr. Costa da Silva

We’re pleased to inform you that your manuscript has been judged scientifically suitable for publication and will be formally accepted for publication once it meets all outstanding technical requirements.

Kind regards,

Raphael Mendonça Guimaraes, PhD

Academic Editor

PLOS ONE

---

## [Editor Report · Acceptance letter]

16 Mar 2024

PONE-D-23-12150R1 

PLOS ONE

Dear Dr. Silva, 

I'm pleased to inform you that your manuscript has been deemed suitable for publication in PLOS ONE. Congratulations! Your manuscript is now being handed over to our production team.

Kind regards, 

on behalf of

Dr. Raphael Mendonça Guimaraes 

Academic Editor

PLOS ONE